# Conformal Prediction for Deep Classifier via Label Ranking

## Abstract

Conformal prediction is a statistical framework that generates prediction sets containing ground-truth labels with a desired coverage guarantee. The predicted probabilities produced by machine learning models are generally miscalibrated, leading to large prediction sets in conformal prediction. In this paper, we empirically and theoretically show that disregarding the probabilities' value will mitigate the undesirable effect of miscalibrated probability values. Then, we propose a novel algorithm named *Sorted Adaptive prediction sets* (SAPS), which discards all the probability values except for the maximum softmax probability. The key idea behind SAPS is to minimize the dependence of the non-conformity score on the probability values while retaining the uncertainty information. In this manner, SAPS can produce sets of small size and communicate instance-wise uncertainty. Theoretically, we provide a finite-sample coverage guarantee of SAPS and show that the expected value of set size from SAPS is always smaller than APS. Extensive experiments validate that SAPS not only lessens the prediction sets but also broadly enhances the conditional coverage rate and adaptation of prediction sets.

## 1 Introduction

Machine learning is being deployed in many high-stakes tasks, such as autonomous driving (Bojarski et al., 2016), medical diagnostics (Caruana et al., 2015) and financial decision-making. The trust and safety in these applications are critical, as any erroneous prediction can be costly and dangerous. To assess the reliability of predictions, a popular solution is to quantify the model uncertainty, such as confidence calibration (Guo et al., 2017), MC-Dropout (Gal & Ghahramani, 2016), and Bayesian neural network (Smith, 2013; Blundell et al., 2015). However, these methods lack theoretical guarantees of model performance. This gives rise to the importance of Conformal Prediction (CP) (Vovk et al., 2005; Shafer & Vovk, 2008; Balasubramanian et al., 2014; Angelopoulos & Bates, 2021), which yields prediction sets containing ground-truth labels with a desired coverage guarantee.

In the literature, CP algorithms design non-conformity scores to quantify the degree of deviation between a new instance and the training data, determining the size of the final prediction sets. A higher non-conformity score is associated with a larger prediction set or region, indicating a lower level of confidence in the prediction. For example, Adaptive Prediction Sets (APS) (Romano et al., 2020) calculates the score by accumulating the sorted softmax values in descending order. However, the softmax probabilities typically exhibit a long-tailed distribution, allowing for easy inclusion of those tail classes in the prediction sets. To alleviate this issue, Regularized Adaptive Prediction Sets (RAPS) (Angelopoulos et al., 2021b) exclude unlikely classes by appending a penalty to classes beyond some specified threshold. The non-conformity score of RAPS still involves in unreliable softmax probabilities, leading to suboptimal performance in conformal prediction. This motivates our question: *does the probability value play a critical role in conformal prediction?*

In this work, we show that the value of softmax probability might be redundant information for constructing the non-conformity score in conformal prediction. We provide an empirical analysis by removing the exact value of softmax probability while preserving the relative rankings of labels. The results indicate that APS using label ranking yields much smaller prediction sets than APS using the softmax outputs, at the same coverage rate. Theoretically, we show that, by removing the probability value, the size of prediction sets generated by APS is consistent with model prediction accuracy. In

other words, a model with higher accuracy can produce smaller prediction sets, using APS without access to the probability value. The details of the analysis are presented in Subsection 3.1.

Inspired by the analysis, our key idea is to minimize the dependence of the non-conformity score on the probability values, while retaining the uncertainty information. Specifically, we propose *Sorted Adaptive prediction sets* (dubbed **SAPS**), which discards all the probability values except for the maximum softmax probability in the construction of non-conformity score. This can be achieved by replacing the non-maximum probability values with a constant, after sorting in descending order. In effect, SAPS can not only produce sets of small size but also communicate instance-wise uncertainty. Theoretically, we show that the expected value of set size from SAPS is always smaller than APS, using a well-calibrated model.

To verify the effectiveness of our method, we conduct thorough empirical evaluations on common benchmarks, including CIFAR-10, CIFAR-100 (Krizhevsky et al., 2009), and ImageNet (Deng et al., 2009). The results demonstrate that SAPS achieves superior performance over the compared methods, including APS and RAPS. For example, our approach reduces the average size of prediction sets from 20.95 to 2.98 – only $\frac{1}{7}$ of the prediction set size from APS. Compared to RAPS, we show that SAPS not only produces a higher conditional coverage rate but also exhibits better adaptability to the instance difficulty.

We summarize our contributions as follows:

1. We empirically show that the probability value is not necessary in APS. Specifically, APS without probability value generates smaller prediction sets than vanilla APS. Moreover, we theoretically show that APS without probability value can provide stable prediction sets, in which the set size is consistent with the prediction accuracy of models.

2. We propose a novel non-conformity score–SAPS that minimizes the dependency on probability value while retaining the uncertainty information. We provide theoretical analyses to show the marginal coverage properties of SAPS and the advantage over APS.

3. Extensive experimental results demonstrate the effectiveness of our proposed method. We show that SAPS not only lessens the prediction sets but also broadly enhances the conditional coverage rate and adaptation of prediction sets.

4. We provide analyses to improve our understanding of the proposed method. In particular, we contrast with a special variant of RAPS and demonstrate the advantages of our method. We also investigate the effect of calibration on our method.

## 2 PRELIMINARIES

In this work, we consider the multi-class classification task with $K$ classes. Let $\mathcal{X} \subset \mathbb{R}^d$ be the input space and $\mathcal{Y} := \{1, \ldots, K\}$ be the label space. We use $\hat{\pi} : \mathcal{X} \to \mathbb{R}^K$ to denote the pre-trained neural network that is used to predict the label of a test instance. Let $(X, Y) \sim \mathcal{P}_{\mathcal{X}\mathcal{Y}}$ denote a random data pair satisfying a joint data distribution $\mathcal{P}_{\mathcal{X}\mathcal{Y}}$. Ideally, $\hat{\pi}_y(\boldsymbol{x})$ can be used to approximate the conditional probability of class $i$ given feature $\boldsymbol{x}$, i.e., $\mathbb{P}[Y = y | X = \boldsymbol{x}]$. Then, the model prediction in classification tasks is generally made as: $\hat{y} = \arg\max_{y \in \mathcal{Y}} \hat{\pi}_y(\boldsymbol{x})$.

**Conformal prediction.** To provide a formal guarantee for the model performance, conformal prediction (Vovk et al., 2005) is designed to produce prediction sets containing ground-truth labels with a desired probability. Instead of predicting one-hot labels from the model outputs, the goal of conformal prediction is to construct a set-valued mapping $\mathcal{C} : \mathcal{X} \to 2^{\mathcal{Y}}$, which satisfies the *marginal coverage*:

$$\mathbb{P}(Y \in \mathcal{C}_{1-\alpha}(X)) \geq 1 - \alpha, \tag{1}$$

where $\alpha \in (0, 1)$ denotes the desired error rate and $\mathcal{C}_{1-\alpha}(X)$ is a subset of $\mathcal{Y}$. Particularly, a smaller value of $\alpha$ will enlarge the predictions set, i.e.,

$$\alpha_1 > \alpha_2 \implies \mathcal{C}_{1-\alpha_1}(X) \subseteq \mathcal{C}_{1-\alpha_2}(X) \tag{2}$$

Before deployment, conformal prediction begins with a calibration step, using a calibration set $\mathcal{D}_{cal} := \{(\boldsymbol{x}_i, y_i)\}_{i=1}^n$. The data of the calibration set is also i.i.d. drawn from the distribution

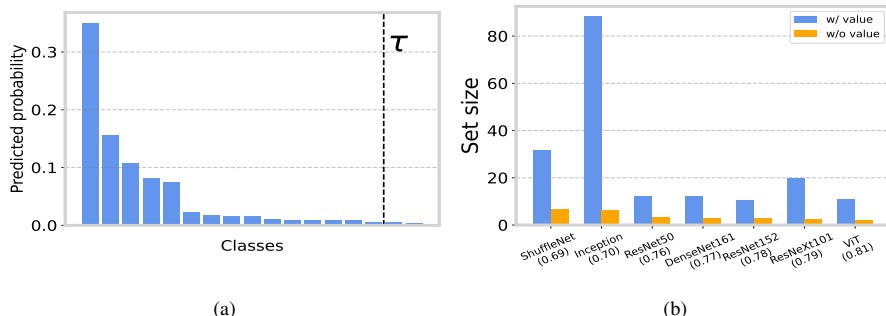

(a)             (b)

Figure 1: (a) Sorted softmax probabilities of an example from ImageNet in descending order. (b) Set size for APS on various models. We use "w/ value" and "w/o value" to represent the vanilla APS and APS with label ranking, respectively. The numbers in brackets represent the prediction accuracy of the model. The sizes of the prediction sets are small after removing the probability value.

$\mathcal{P_{XY}}$. Specifically, we calculate a non-conformity score $s_i = S(\boldsymbol{x}_i, y_i)$ for each example $(\boldsymbol{x}_i, y_i)$ in the calibration set, where $s_i$ measures the degree of deviation between the given example and the training data. The $1 - \alpha$ quantile of the non-conformity scores $\{s_i\}_{i=1}^n$ is then determined as a threshold $\tau$. Formally, the value of $\tau$ can be obtained as shown below:

$$\tau = \inf\{s : \frac{|\{i \in \{1, \ldots, n\} : s_i \leq s\}|}{n} \geq \frac{\lceil (n+1)(1-\alpha) \rceil}{n}\}$$

During testing, we calculate the non-conformity score for each label given a new instance $\boldsymbol{x}_{n+1}$. Then, the corresponding prediction set $\mathcal{C}(\boldsymbol{x}_{n+1})$ comprises possible labels whose non-conformity score $S(\boldsymbol{x}_{n+1}, y)$ falls within the threshold $\tau$:

$$\mathcal{C}_{1-\alpha}(\boldsymbol{x}_{n+1}) = \{y \in \mathcal{Y} : S(\boldsymbol{x}_{n+1}, y) \leq \tau\}. \tag{3}$$

The equation above exhibits a nesting property of threshold, i.e., $\tau_1 \leq \tau_2 \implies \{y \in \mathcal{Y} : S(\boldsymbol{x}_{n+1}, y) \leq \tau_1\} \subseteq \{y \in \mathcal{Y} : S(\boldsymbol{x}_{n+1}, y) \leq \tau_2\}$. With a lower value of $\tau$, the model tends to produce a smaller prediction set, indicating a higher level of confidence in the prediction. Conversely, the increase of $\tau$ will enlarge the size of the prediction set, suggesting greater uncertainty of the prediction. In this manner, conformal prediction can be used to estimate the uncertainty or reliability of the model's predictions.

**Adaptive prediction sets (APS).** In the APS method (Romano et al., 2020), the non-conformity scores are calculated by accumulating softmax probabilities in descending order. Formally, given a data pair $(\boldsymbol{x}, y)$, the non-conformity score can be computed by:

$$S(\boldsymbol{x}, y, u; \hat{\pi}) := \sum_{i=1}^{o(y, \hat{\pi}(\boldsymbol{x}))-1} \hat{\pi}_{(i)}(\boldsymbol{x}) + u \cdot \hat{\pi}_{(o(y, \hat{\pi}(\boldsymbol{x})))}(\boldsymbol{x}), \tag{4}$$

where $o(y, \hat{\pi}(\boldsymbol{x}))$ denotes the index of $\hat{\pi}_y(\boldsymbol{x})$ in the sorted softmax probabilities, i.e., $\hat{\pi}_{(1)}(\boldsymbol{x}), \ldots, \hat{\pi}_{(K)}(\boldsymbol{x})$, and $u$ is an independent random variable satisfying a uniform distribution on $[0, 1]$. Given a test point $\boldsymbol{x}_{n+1}$, the prediction set of APS with the error rate $\alpha$ is given by $\mathcal{C}_{1-\alpha}(\boldsymbol{x}_{n+1}, u_{n+1}) := \{y \in \mathcal{Y} : S(\boldsymbol{x}_{n+1}, y, u_{n+1}; \hat{\pi}) \leq \tau\}$. With the non-conformity score in Eq. 4, APS achieves a finite-sample marginal coverage guarantee. However, the softmax probabilities $\hat{\pi}(\boldsymbol{x})$ typically exhibit a long-tailed distribution, where the tail probabilities with small values can be easily included in the prediction sets. Consequently, APS tends to produce large prediction sets for all inputs, regardless of the instance difficulty. For example, in Figure 1a, the long-tail probability distribution results in the non-conformity scores of many classes falling within $\tau$. This motivates our analysis to investigate the role of probability value in conformal prediction.

## 3 MOTIVATION AND METHOD

### 3.1 MOTIVATION

To analyze the role of probability values, we perform an ablation study by removing the influence of probability values in Eq. 4. In particular, we replace these probabilities with a constant $\gamma$ (e.g., $\gamma = 1$), after sorting in descending order. With the constant $\gamma$, the modified non-conformity score for a data pair $(\boldsymbol{x}, y)$ with a pre-trained model $\hat{\pi}$ is:

$$S(\boldsymbol{x}, y, u; \hat{\pi}) := \gamma \cdot [o(y, \hat{\pi}(\boldsymbol{x})) - 1 + u].\tag{5}$$

In the analysis, we fix the constant as 1 for simplification. Then, we conduct experiments on ImageNet (Deng et al., 2009) to compare the new non-conformity score to the vanilla APS. Here, we set the desired error rate as 10%, i.e., $\alpha = 0.1$. Following previous works (Romano et al., 2019; Angelopoulos et al., 2021b; Ghosh et al., 2023), we first randomly split the test dataset of ImageNet into two subsets: a conformal calibration subset of size 30K and a test subset of size 20K. For network architecture, we use seven models trained on ImageNet, with different levels of prediction performance (see Figure 1b). All models are calibrated by the temperature scaling procedure (Guo et al., 2017). Finally, experiments are repeated ten times and the median results are reported.

**Probability values are not necessary.** Figure 1b presents the results on various models, using APS with/without the probability value. The results indicate that APS solely based on label ranking generates smaller prediction sets than those generated with the vanilla APS, across various models. For example, with the Inception model, removing the probability values reduces the set size of 88.18 to 6.33. Using a transformer-based ViT model (Touvron et al., 2021), APS without probability value also obtains a smaller set size. From the comparison, we show that the probability value might be redundant information for non-conformity scores in conformal prediction. We proceed by theoretically analyzing the advantage of removing probability values in APS.

**A theoretical interpretation.** The empirical results above demonstrate that the probability value is not a critical component of the non-conformity score for conformal prediction. Here, we provide a formal analysis of APS without probability value through the following theorem:

**Theorem 1.** *Let $A_r$ denote the accuracy of the top $r$ predictions on a trained model $\hat{\pi}$. Given a significance level $\alpha$, for any test instance $\boldsymbol{x} \sim \mathcal{P}_{\mathcal{X}}$ and an independent random variable $u \sim U[0, 1]$, if there exists a number $k$ satisfying $A_k \geq 1 - \alpha > A_{k-1}$, the size of prediction set $\mathcal{C}_{1-\alpha}(\boldsymbol{x}, u)$ generated by APS without probability value can be obtained by*

$$|\mathcal{C}_{1-\alpha}(\boldsymbol{x}, u)| = \begin{cases} k, & \text{if } u < \dfrac{1 - \alpha - A_{k-1}}{A_k - A_{k-1}}, \\ k - 1, & \text{otherwise.} \end{cases}\tag{6}$$

*The expected value of the set size can be given by*

$$\mathbb{E}_{u \sim [0,1]}[\mathcal{C}_{1-\alpha}(\boldsymbol{x}, u)] = k - 1 + \frac{1 - \alpha - A_{k-1}}{A_k - A_{k-1}}.\tag{7}$$

The proof of Theorem 1 can be found in Appendix A. As indicated by Eq. 7, the prediction set size generated by APS without probability value is consistent with $k$. In other words, a higher model accuracy will lead to a smaller value of $k$, indicating a smaller prediction sets. This argument is clearly supported by experimental results shown in Figure 1b. In particular, we observe that using APS without probability value, models with higher accuracy produce a smaller prediction sets, while the vanilla APS does not exhibit this characteristic. For example, using ResNeXt101, the model achieves higher prediction accuracy than using ResNet152, while producing a larger prediction set. The analysis demonstrates the advantage of removing probability value in APS, via decreasing the sensitivity to tail probabilities.

### 3.2 METHOD

In the analysis above, we demonstrate that removing the probability value in APS can largely decrease the size of prediction sets. On the other hand, the expected value of the set size (shown in

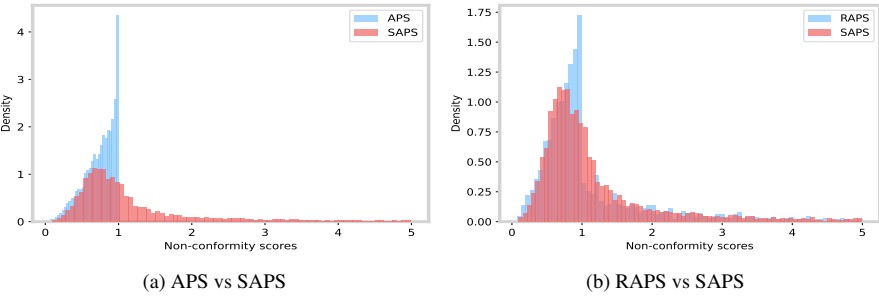

(a) APS vs SAPS

(b) RAPS vs SAPS

Figure 2: Distribution of non-conformity scores for examples with wrong predictions.

Eq. 6) will oscillate between $k - 1$ and $k$, after removing the probability value. This implies a shortcoming of the modified non-conformity score in adaptation to instance-wise uncertainty, which may cause overcovering on easy examples.

To alleviate this limitation, we propose a novel conformal prediction algorithm, named *Sorted Adaptive Prediction Sets*. The key idea behind this algorithm is to minimize the dependence of the non-conformity score on the probability values while retaining the uncertainty information. In particular, we discard all the probability values except for the maximum softmax probability, which is usually used to measure the model confidence in the prediction. Formally, the non-conformity score can be calculated as

$$S(\boldsymbol{x}, y, u; \hat{\pi}) := \begin{cases} u \cdot \hat{\pi}_{max}(\boldsymbol{x}), & \text{if} \quad o(y, \hat{\pi}(\boldsymbol{x})) = 1, \\ \hat{\pi}_{max}(\boldsymbol{x}) + (o(y, \hat{\pi}(\boldsymbol{x})) - 2 + u) \cdot \lambda, & \text{otherwise}, \end{cases} \quad (8)$$

where $\lambda$ is a hyperparameter showing the weight of ranking information, $\hat{\pi}_{max}(\boldsymbol{x})$ denotes the maximum softmax probability and $u$ denotes a uniform random variable. We provide a detailed analysis on the effect of $\lambda$ in Section 5.

In Eq. 8, we incorporate the uncertainty information via the maximum probability $\hat{\pi}_{max}(\boldsymbol{x})$, and use the constant $\lambda$ to mitigate the undesirable influence of tail probabilities. In this manner, the SAPS method can not only produce sets of small size, but also communicate instance-wise uncertainty. In other words, the prediction set can be smaller for easy inputs than for hard ones. We illustrate with an experiment in Figure 2, where the examples with wrong predictions have higher non-conformity scores provided by SAPS, compared to those of APS and RAPS. Moreover, for examples with correct predictions, the non-conformity scores defined in APS, RAPS, and APS are equivalent as the ranks of ground-truth labels are 1 (i.e., $S(\boldsymbol{x}, y, u; \hat{\pi}) = u \cdot \hat{\pi}_{max}(\boldsymbol{x})$). The results indicate that the non-conformity score of SAPS can better characterize the deviation between a given example and the training data.

In what follows, we provide a formal analysis to show the effectiveness of our SAPS algorithm. We start by showing the finite-sample marginal coverage properties:

**Proposition 1.** *(Coverage guarantee of SAPS). Suppose $(\boldsymbol{x}_i, y_i, u_i)_{i=1,\dots,n}$ and $(\boldsymbol{x}_{n+1}, y_{n+1}, u_{n+1})$ are i.i.d. and let the prediction set of SAPS with error rate $\alpha$ as $\mathcal{C}_{1-\alpha}(\boldsymbol{x}, u) := \{y \in \mathcal{Y} : S(\boldsymbol{x}, y, u; \hat{\pi}) \leq \tau\}$, where $S(\boldsymbol{x}, y, u; \hat{\pi})$ is the score function defined as in Eq. 8. Then for $\tau$ defined as $1 - \alpha$ quantile of scores $\{S(\boldsymbol{x}_i, y_i, u_i; \hat{\pi})\}_{i=1,\dots,n}$, we have the coverage guarantee:*

$$\mathbb{P}\left(y_{n+1} \in \mathcal{C}_{1-\alpha}\left(\boldsymbol{x}_{n+1}, u_{n+1}\right)\right) \geq 1 - \alpha$$

The corresponding proof is provided in Appendix B. In the following, we further prove that SAPS always dominates APS in the size of prediction sets.

**Proposition 2.** *(SAPS dominates APS) If $\hat{\pi}$ is well-calibrated and $\lambda \geq 1 - \frac{1}{K}$, for any test instance $\boldsymbol{x} \sim \mathcal{P}_{\mathcal{X}}$ with a significance level $\alpha$, we have*

$$\mathbb{E}_{u \sim [0,1]}\{|\mathcal{C}(\boldsymbol{x}, u)|\} \leq \mathbb{E}_{u \sim [0,1]}\{|\tilde{\mathcal{C}}(\boldsymbol{x}, u)|\},$$

*where $u \sim U[0, 1]$. $\mathcal{C}(\cdot)$ and $\tilde{\mathcal{C}}(\cdot)$ represent the prediction set from SAPS and APS, respectively.*

In other words, SAPS consistently generates a smaller prediction set than APS when the oracle model is available, while both algorithms maintain the desired marginal coverage rate. The formal pseudocode for SAPS is provided in the Appendix H.

# 4 EXPERIMENTS

## 4.1 EXPERIMENTAL SETUP

**Classification datasets.** We consider three prominent datasets in our study: ImageNet (Deng et al., 2009), CIFAR-100 and CIFAR-10 (Krizhevsky et al., 2009), which are common benchmarks for conformal prediction. In the case of ImageNet, we split the test dataset containing 50000 images into 30000 images for the calibration set and 20000 images for the test set. For CIFAR-100 and CIFAR-10, We divide the corresponding test dataset equally into a calibration set containing 5000 images and a test set containing 5000 images.

**Models.** We employ twelve different classifiers, including nine standard classifiers, two transformer-based models, i.e., ViT (Dosovitskiy et al., 2020) and DeiT (Touvron et al., 2021), and a Vision-Language Model named CLIP (Radford et al., 2021). Aside from CLIP with zero-shot prediction capabilities, the remaining models are the pre-trained models on ImageNet. For CIFAR-10 and CIFAR-100, these models will be fine-tuned on the pre-trained models. Moreover, all classifiers are calibrated by the Temperature scaling procedure (Guo et al., 2017).

**Conformal prediction algorithms.** We compare the proposed method against APS (Romano et al., 2020) and RAPS (Angelopoulos et al., 2021b). Then, we choose the hyper-parameter that achieves the smallest set size on a validation set, which is a subset of the calibration set. Specifically, we tune the regularization hyperparameter of RAPS in $\{0.001, 0.01, 0.1, 0.15, \ldots, 0.5\}$ and hyperparameter $\lambda$ in $\{0.02, 0.05, 0.1, 0.15, \ldots, 0.6\}$ for SAPS. All experiments are conducted with ten trials, and the median results are reported.

**Evaluation.** The primary metrics used for the evaluation of prediction sets are set size (average length of prediction sets; small value means high efficiency) and marginal coverage rate (fraction of testing examples for which prediction sets contain the ground-truth labels). These two metrics can be formally represented as :

$$\text{Size} = \frac{1}{N_{test}} \sum_{i=1}^{N_{test}} |\mathcal{C}(\boldsymbol{x}_i)|$$

$$\text{coverage rate} = \frac{1}{N_{test}} \sum_{i=1}^{N_{test}} \mathbf{1}(y_i \in \mathcal{C}(\boldsymbol{x}_i))$$

*Conditional coverage rate.* In this work, we propose an alternative metric to the SSCV criterion named Each-Size Coverage Violation (ESCV) that can be utilized for any number of classes, as shown below:

$$\text{ESCV}(\mathcal{C}, K) = \sup_j \max(0, (1 - \alpha) - \frac{|\{i \in \mathcal{J}_j : y_i \in \mathcal{C}(\boldsymbol{x}_i)\}|}{|\mathcal{J}_j|})$$

where $\mathcal{J}_j = \{i : |\mathcal{C}(\boldsymbol{x}_i)| = j\}$ and $j \in \{1, \ldots, K\}$. Specifically, ESCV measures the most significant violation of prediction sets with each size. This metric is practical because it only requires the set size, and is suitable for any classification problem, spanning from binary classes to large classes.

## 4.2 RESULTS

**SAPS generates smaller prediction sets.** In Table 1, the performance of set sizes and coverage rates for various classification tasks are presented. We can observe that the coverage rate of all conformal prediction methods is close to the desired coverage $1 - \alpha$. At different significance levels (i.e., 0.1 and 0.05), the prediction set size is consistently reduced by SAPS for ImageNet, CIFAR-100 and CIFAR-10, compared to APS and RAPS. For example, when evaluated on ImageNet, SAPS reduces the average set size from 20.95 of APS to 2.98. Moreover, as the scale of the classification task increases, the efficiency improvement achieved by SAPS becomes increasingly evident. Overall, the experiments show that our method has the desired coverage rate and a smaller set size than APS and RAPS. Due to space constraints, we only report the average results of multiple models on various classification tasks in Table 1, and detailed results for each model are available in Appendix D.

Table 1: Results of average set sizes on different datasets. We evaluate the performance of SAPS, APS, and RAPS by calculating the average set size across multiple models. It is evident that SAPS consistently outperforms APS and RAPS in various classification tasks, such as ImageNet, CIFAR-100, and CIFAR-10, and different significance levels ($\alpha = 0.1, 0.05$). **Bold** numbers indicate optimal performance.

| | $\alpha = 0.1$ | | | | | | $\alpha = 0.05$ | | | | | |
| | Coverage | | | Size ↓ | | | Coverage | | | Size ↓ | | |
| Datasets | APS | RAPS | SAPS | APS | RAPS | SAPS | APS | RAPS | SAPS | APS | RAPS | SAPS |
|---|---|---|---|---|---|---|---|---|---|---|---|---|
| ImageNet | 0.899 | 0.900 | 0.900 | 20.95 | 3.29 | **2.98** | 0.949 | 0.950 | 0.950 | 44.67 | 8.57 | **7.55** |
| CIFAR-100 | 0.899 | 0.900 | 0.899 | 7.88 | 2.99 | **2.67** | 0.950 | 0.949 | 0.949 | 13.74 | 6.42 | **5.53** |
| CIFAR-10 | 0.899 | 0.900 | 0.898 | 1.97 | 1.79 | **1.63** | 0.950 | 0.950 | 0.950 | 2.54 | 2.39 | **2.25** |

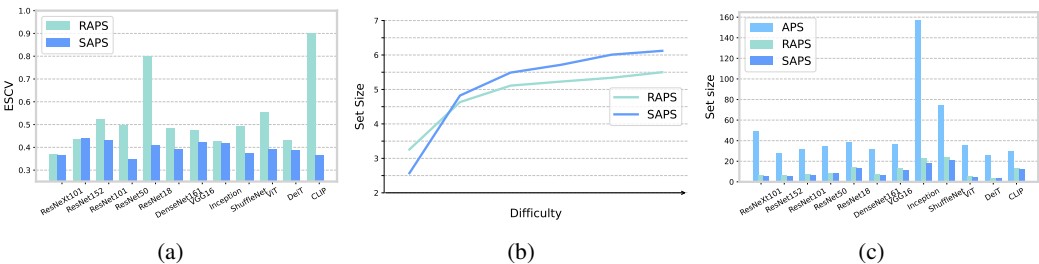

(a)  (b)  (c)

Figure 3: (a) ESCV with different models on ImageNet with $\alpha = 0.1$. A good conformal prediction algorithm should keep the y-axis (e.g., ESCV) small. The results show that SAPS outperforms RAPS on most models. (b) Set size under different difficulties on VGG16. Small sets are required for easy examples, while hard ones require large sets. For example, SAPS generates smaller sets than RAPS on easy examples, but with difficulty improving, the size of SAPS will be larger than SAPS. (c) Set size on ImageNet-V2 at $\alpha = 0.1$.

**SAPS acquires lower conditional coverage violation.** In Figure 3a, we demonstrate that SAPS not only outperforms in efficiency but also boosts the conditional coverage rate, i.e., ESCV. Given that our study primarily focuses on improving the efficiency of prediction sets, the comparison of ESCV is limited to SAPS and RAPS. The results, shown in Figure 3a, demonstrate that for most models, SAPS would get smaller ESCV than RAPS. For example, on CLIP, SAPS reduces the ESCV from $0.9$ to $0.37$. In addition, on ImageNet, we can observe that the ESCV of SAPS for different models is more stable than RAPS. Specifically, the ESCV of SAPS can keep a low value on most models, but in the case of RAPS, the maximum ESCV even gets $0.9$. The detailed results on CIFAR-10 and CIFAR-100 are provided in Appendix E.

**SAPS exhibits higher adaptation.** Adaptation indicates the ability to adjust the size of the prediction set based on the complexity or difficulty of individual examples. In other words, the prediction sets should be small for easy examples but large for hard ones. In this work, we employ the rank of the ground-truth labels in the sorted softmax probabilities to denote the difficulty. For instance, examples with serious difficulty are assigned high ranks for their ground-truth labels. In Figure 3b, the results show that the set size of SAPS has higher adaptation. Specifically, compared with RAPS, SAPS produces smaller sets for accurate predictions but larger sets for hard examples on VGG16. More results of different models are reported in Appendix F. Overall, We show that SAPS can improve the adaptation of prediction sets while maintaining small set sizes.

**Experiments on distribution shifts.** We also verify the effectiveness of our method on the new distribution, which is different from the training data distribution. Specifically, We divide the test dataset of ImageNet-V2 (Recht et al., 2019), which exhibits a distribution shift compared to the ImageNet, equally into a calibration set containing 5000 images and a test set containing 5000 images. Then, the test models are only pre-trained on ImageNet and not be fine-tuned. As shown in Figure 3c, the result shows that under $\alpha = 0.1$, our method can also generate the smallest sets when the conformal calibration set and the test set come from a new distribution.

Table 2: Set size and ESCV for RAPS ($k_r = 1$) and SAPS. We report the average value across various models with $\alpha = 0.1$. The detailed results of each model are provided in the Appendix G. **Bold** numbers indicate optimal performance.

| Datasets | Coverage | | Size ↓ | | ESCV ↓ | |
|---|---|---|---|---|---|---|
| | RAPS($k_r = 1$) | SAPS | RAPS($k_r = 1$) | SAPS | RAPS($k_r = 1$) | SAPS |
| ImageNet | 0.900 | 0.900 | 3.24 | **2.98** | 0.631 | **0.396** |
| CIFAR-100 | 0.899 | 0.899 | 2.79 | **2.67** | 0.390 | **0.302** |
| CIFAR-10 | 0.900 | 0.898 | **1.62** | 1.63 | 0.138 | **0.089** |

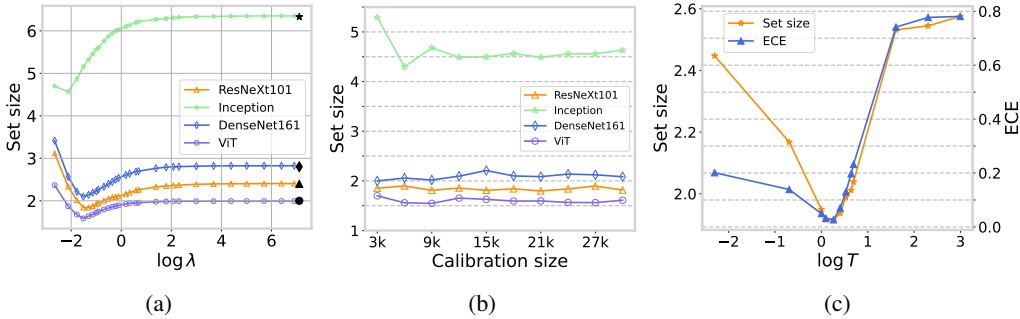

(a)                              (b)                              (c)

Figure 4: (a) Effect of the $\lambda$ on set size across various models. The black markers ($\star, \blacklozenge, \blacktriangle, \bullet$) represent the results of APS without probability value. (b) Effect of the calibration dataset size on set size across various models. (c) Relationship between temperature and the set size of SAPS on ResNet152, where the horizon axis represents the log transformation of temperature $T$.

## 5 DISCUSSION

**Effect of $\lambda$ and calibration size.** In SAPS, we choose an optimal $\lambda$ by performing a search over an sequence to minimize the set size on a validation set. In this work, the validation set constitutes 20% of the calibration set. Here, we provide an empirical analysis to show *whether set size is sensitive to $\lambda$ and calibration size.* To this end, we conduct two experiments on ImageNet to analyze the effects of $\lambda$ and the size of the calibration set.

We present the results of four models in Figure 4. Indeed, Figure 4a illustrates that one can efficiently utilize grid search to find the optimal $\lambda$. Furthermore, as depicted in Figure 4b, nearly all models maintain stable results when the number of calibration sets increases. Overall, the results demonstrate that the set size is not sensitive to variations in $\lambda$ and calibration size.

**SAPS vs. RAPS ($k_r = 1$).** While SAPS has demonstrated strong promise, it shares a similarity in the definition of non-conformity scores with RAPS ($k_r = 1$), as shown below:

$$S(\boldsymbol{x}, y, u, \hat{\pi}) = \sum_{i=1}^{o(y,\hat{\pi}(\boldsymbol{x}))-1} \hat{\pi}_{(i)}(\boldsymbol{x}) + u * \hat{\pi}_{o(y,\hat{\pi}(\boldsymbol{x}))}(\boldsymbol{x}) + \phi \cdot (o(y, \hat{\pi}(\boldsymbol{x})) - k_r)^+.$$

Here, $\phi$ represents the weight of regularization and $(z)^+$ denotes the positive part of $z$. To this end, we conduct a comprehensive experiment with $\alpha = 0.1$ on CIFAR-10, CIFAR-100, and ImageNet to compare SAPS and RAPS ($k_r = 1$).

As indicated in Table 2, SAPS outperforms RAPS ($k_r = 1$) in large-scale classification scenarios, achieving smaller prediction sets and lower conditional coverage violations. In the small-scale classification task (i.e., CIFAR-10), SAPS produces a comparable set size with RAPS ($k_r = 1$), and the ESCV of SAPS was more than 1.5 times as small as those from RAPS. Overall, employing a constant to substitute the noisy probabilities is an effective way to alleviate the negative implications of noisy probabilities further.

**Relation to temperature scaling.** In the literature, temperature scaling calibrates softmax probabilities output by models by minimizing Expected Calibration Error (ECE), leading to a reliable maximum probability. As defined in Eq. 8, the remaining probability value used in the non-conformity scores is the maximum probability, a question arises: *what is a relation between temperature scaling and the set size of SAPS?* Here, we vary the value of temperature $T = \{0.1, 0.5, 1, 1.1, 1.3, 1.5, 1.7, 1.9, 2, 5, 10, 20\}$ in temperature scaling. We utilize SAPS to test the ResNet152 model calibrated by different temperatures on the ImageNet benchmark. The results indicate that there exists a consistency between the temperature value and the set size.

As illustrated in Figure 4c, the temperature with the lowest ECE can achieve the smallest prediction sets. Specifically, the optimal temperature on ECE and set size are the same, i.e., 1.3. Moreover, as the ECE increases, the set size also increases. Indeed, temperature scaling can not change the permutation of the softmax probabilities but improves the confidence level of the maximum probability, resulting in the non-conformity scores of SAPS being more reliable. Overall, for SAPS, better confidence calibration can produce smaller prediction sets.

## 6 RELATED WORK

Conformal prediction is a statistical framework characterized by a finite-sample coverage guarantee. It has been utilized in various tasks including regression (Lei & Wasserman, 2014; Romano et al., 2019), classification (Sadinle et al., 2019), structured prediction (Bates et al., 2021), Large-Language Model (Kumar et al., 2023; Ren et al., 2023) and so on.

The primary focal points of CP are reducing prediction set size and enhancing coverage rate. Strategies to reduce the set size can be roughly split into the following two branches. The first approach involves leveraging post-hoc technologies (Romano et al., 2020; Angelopoulos et al., 2021a; Ghosh et al., 2023). There exist others concentrate on unique settings such as federated learning (Lu et al., 2023) and multi-label problem (Cauchois et al., 2020; Fisch et al., 2022; Papadopoulos, 2014), outlier detection (Bates et al., 2023; Chen et al., 2023; Guan & Tibshirani, 2022). Most existing post-hoc methods construct the non-conformity score based on unreliable probability values, leading to suboptimal performance. Different from previous post-hoc methods, we show that probability value is not necessary in non-conformity scores and design an effective method to remove the probability value while retaining uncertainty information.

Another avenue of research focuses on developing new training algorithms to reduce the average prediction set size (Colombo & Vovk, 2020; Chen et al., 2021; Stutz et al., 2022; Einbinder et al., 2022b; Bai et al., 2022; Fisch et al., 2021). Those training methods are usually computationally expensive due to the model retraining. Additionally, there is a growing number of work dedicated to enhancing coverage rate (Vovk, 2012; Shi et al., 2013; Löfström et al., 2015; Ding et al., 2023), including efforts to maintain the marginal coverage rate by modifying the assumption of exchangeability to accommodate factors such as adversaries (Gendler et al., 2021), covariate shifts (Tibshirani et al., 2019), label shifts (Podkopaev & Ramdas, 2021) and noisy labels (Einbinder et al., 2022a; Sesia et al., 2023). In this study, SAPS not only lessens the prediction sets, but also broadly enhances the conditional coverage rate and adaptation of prediction sets.

## 7 CONCLUSION

In this paper, we present SAPS, a simple alternative CP algorithm that generates smaller prediction sets. By integrating the label rank, SAPS effectively mitigates the negative effect of small probability, resulting in a stable prediction set. The extensive experiments show that SAPS can improve the conditional coverage rate and adaptation while maintaining a small prediction set. This method can be easily applied to any pre-trained classifiers. We hope that our insights inspire future research to leverage label ranking information for conformal prediction.

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

## A    PROOF OF THEOREM 1

*Proof.* For simplicity, let $a_r = A_r - A_{r-1}$ for $r \geq 1$ and $A_0 = 0$. $a_r$ represents the rate of examples in the calibration set whose ground-truth labels' rank is $r$. The non-conformity scores of examples whose rank of ground-truth labels are $r$ can be computed by

$$S(\boldsymbol{x}, y, u; \hat{\pi}) = r - u,$$

where $u \sim U[0, 1]$. Thus, we can observe that the scores of these examples are uniformly distributed on $[r-1, r]$. Scores of examples whose rank of ground-truth labels are less than or equal to $r$ are no more than $r$. In other words, the rate of examples in the calibration set with scores lower than $r$ is $A_r$. Given that $k$ satisfies $A_k \geq 1 - \alpha > A_{k-1}$, the $1 - \alpha$ quantile of scores for the calibration set, i.e., the calibrated threshold $\tau$, is located in the interval $[k-1, k]$. Since the scores in the interval $[k-1, k]$ are uniformly distributed and the rate of examples whose scores in $[k-1, k]$ is $a_k$, $\tau$ is equivalent to the $\frac{1-\alpha-A_{k-1}}{a_k}$ quantile of $[k-1, k]$. Formally, we can get the value of $\tau$ by

$$\tau = (k-1) + \frac{1 - \alpha - A_{k-1}}{a_k}.$$

Based on the definition of prediction set in Eq. 3, the prediction set of a test example $\boldsymbol{x}$ is equal to

$$\mathcal{C}_{1-\alpha}(\boldsymbol{x}, u) = \{y \in \mathcal{Y} : o(y, \hat{\pi}(\boldsymbol{x})) - u \leq \tau\}. \tag{9}$$

Next, we analyze the prediction set $\mathcal{C}_{1-\alpha}(\boldsymbol{x}, u)$. When $y \in \mathcal{Y}$ satisfies the inequation $o(y, \hat{\pi}(\boldsymbol{x})) \leq k-1$, $S(\boldsymbol{x}, y, u; \hat{\pi})$ must be smaller than $\tau$. Thus, the size of sets is at least $k-1$. But for $y$ satisfying $o(y, \hat{\pi}(\boldsymbol{x})) = k$, we can have

$$o(y, \hat{\pi}(\boldsymbol{x})) - u \leq \tau$$
$$k - u \leq (k-1) + \frac{1 - \alpha - A_{k-1}}{a_k}$$
$$1 - \frac{1 - \alpha - A_{k-1}}{a_k} \leq u$$

Thus, there exists a probability of $\frac{1-\alpha-A_{k-1}}{a_k}$ such that $S(\boldsymbol{x}, y, u; \hat{\pi}) \leq \tau$. Finally, the expected value of the set size for the test example $\boldsymbol{x}$ is

$$\mathbb{E}_{u \sim U}[|\mathcal{C}_{1-\alpha}(\boldsymbol{x}, u)|] = k - 1 + \frac{1 - \alpha - A_{k-1}}{a_k} = k - 1 + \frac{1 - \alpha - A_{k-1}}{A_k - A_{k-1}} \tag{10}$$

$\square$

## B    PROOF OF PROPOSITION 1

We first introduce a coverage guarantee theorem. Let $\mathcal{C}(\boldsymbol{x}, u, \tau) : \mathbb{R}^d \times [0, 1] \times \mathbb{R} \to 2^{\mathcal{Y}}$ denote a set-valued mapping. Then, given a data set $\{(\boldsymbol{x}_i, y_i)\}_{i=1,\ldots,n}$ and independent random variables $\{u_i\}_{i=1,\ldots,n}$, a threshold $\tau_{cal}$ for the significance level $\alpha$ is given by

$$\tau_{cal} = \inf \left\{ \tau : \frac{|\{i : y_i \in \mathcal{C}(\boldsymbol{x}_i, u_i, \tau)\}|}{n} \geq \frac{\lceil (n+1)(1-\alpha) \rceil}{n} \right\}. \tag{11}$$

**Theorem 2.** *(Conformal calibration coverage guarantee,(Angelopoulos et al., 2021b)). Suppose $(\boldsymbol{x}_i, y_i, u_i)_{i=1,\ldots,n}$ and $(\boldsymbol{x}_{n+1}, y_{n+1}, u_{n+1})$ are i.i.d. and let the set-valued mapping $\mathcal{C}(x, u, \tau)$ satisfy a nesting property of $\tau$, i.e., $\tau_1 \leq \tau_2 \implies \mathcal{C}(x, u, \tau_1) \subseteq \mathcal{C}(x, u, \tau_2)$. Then, for $\tau_{cal}$ defined as in Eq. 11, we have the following coverage guarantee:*

$$P(y_{n+1} \in \mathcal{C}(x, u, \tau_{cal})) \geq 1 - \alpha$$

Then, we prove Proposition 1.

*Proof.* Let $\mathcal{C}_{1-\alpha}(\boldsymbol{x}, u) := \{y \in \mathcal{Y} : S(\boldsymbol{x}, y, u; \hat{\pi}) \leq \tau\}$ be $\mathcal{C}(\boldsymbol{x}, u, \tau)$ and $\tau_1 \leq \tau_2$. For a data pair $(\boldsymbol{x}, y)$ and $u \sim U[0, 1]$, if $S(\boldsymbol{x}, y, u; \hat{\pi}) \leq \tau_1$, $S(\boldsymbol{x}, y, u; \hat{\pi}) \leq \tau_2$. Then, We can get

$$\{y \in \mathcal{Y} : S(\boldsymbol{x}, y, u; \hat{\pi}) \leq \tau_1\} \subseteq \{y \in \mathcal{Y} : S(\boldsymbol{x}, y, u; \hat{\pi}) \leq \tau_2\}.$$

Thus, SAPS's prediction set $\mathcal{C}_{1-\alpha}(\boldsymbol{x}, u) := \{y \in \mathcal{Y} : S(\boldsymbol{x}, y, u; \hat{\pi}) \leq \tau\}$ satisfies the nesting property of $\tau$.

In addition, since $\mathcal{C}(\boldsymbol{x}, u, \tau)$ is defined as $\{y \in \mathcal{Y} : S(\boldsymbol{x}, y, u; \hat{\pi}) \leq \tau\}$, the $\tau_{cal}$ defined as in Eq. 11 is equal to $1 - \alpha$ quantile of scores $\{S(\boldsymbol{x}_i, y_i, u_i; \hat{\pi})\}_{i=1,\dots,n}$. Therefore, let $\tau$ be $1 - \alpha$ quantile of scores.

Finally, based on Theorem 2, the coverage guarantee of prediciton set $\mathcal{C}_{1-\alpha}(\boldsymbol{x}, u) := \{y \in \mathcal{Y} : S(\boldsymbol{x}, y, u; \hat{\pi}) \leq \tau\}$ on the test example $(\boldsymbol{x}_{n+1}, y_{n+1})$ can be obtained:

$$P\left(y_{n+1} \in \mathcal{C}_{1-\alpha}\left(\boldsymbol{x}_{n+1}, u_{n+1}\right)\right) \geq 1 - \alpha.$$

$\square$

## C  PROOF OF PROPOSITION 2

We first introduce Theorem 3 describing the distribution of non-conformity scores in APS, then prove Lemma 1 for bounds of softmax probability. Finally, we give the proof of Proposition 2.

**Theorem 3.** *(Einbinder et al., 2022b) The distribution of the non-conformity scores $S$ in equation 4 is uniform conditional on $\boldsymbol{x}$ if $\hat{\pi} = \pi$. That is, $\mathbb{P}[\tilde{S}(\boldsymbol{x}, y, u; \pi) \leq \beta \mid X = \boldsymbol{x}] = \beta$ for all $\beta \in (0, 1)$, where $(\boldsymbol{x}, y)$ is a random sample from $\mathcal{P}$, and $u \sim Uniform[0, 1]$ independent of everything else.*

**Lemma 1.** *Given a discrete probability $\{p_i\}_{i=1}^K$ and $p_i > 0, \forall i \in \{1, \dots, K\}$. The ordered probability list is denoted by $p_{(1)} \geq p_{(2)}, \cdots \geq p_{(K)}$. Then, we have that*

$$\frac{1}{K} \leq p_{(1)} \leq 1,$$

*and*

$$0 < p_{(k)} \leq \frac{1}{k}, k \in \{2, \dots, K\}.$$

*Proof.* We have known that $\sum_{i=1}^K p_i = 1$. Supposed that $p_{(1)} < \frac{1}{K}$. We can have that for $k \geq 2$,

$$p_{(k)} \leq p_{(1)} < \frac{1}{K}.$$

Thus,

$$\sum_{i=1}^K p_i < 1.$$

Finally, there exists a contradiction. It is obviously that $p_{(1)} \leq 1$. Thus, We can conclude that $\frac{1}{K} \leq p_{(1)} \leq 1$.

Next, for any $k \geq 2$, supposed that $p_{(k)} > \frac{1}{k}$. Given that for any $\tilde{k} < k$, it holds that $p_{(\tilde{k})} > \frac{1}{k}$. We deduce that

$$\sum_{i=1}^k p_{(i)} > k \cdot \frac{1}{k} = 1.$$

The above equation contradicts that $\sum_{i=1}^k p_{(i)} \leq 1$ for any $k \geq 2$. It is obvious that $0 < p_{(k)}$. Finally, we can conclude that $0 < p_{(k)} \leq \frac{1}{k}$ for any $k \geq 2$. $\square$

Next, we prove Proposition 2.

*Proof.* Let $S_i(\cdot)$ and $\tilde{S}_i(\cdot)$ denote the non-conformity scores of SAPS and APS, respectively. Since $\hat{\pi}$ is well-calibrated, i.e., $\hat{\pi} = \pi$, for any point $(\boldsymbol{x}_i, y_i)$ from calibration set, $S_i(\cdot)$ and $\tilde{S}_i(\cdot)$ are reduced to $u \cdot \pi_{(1)}(\boldsymbol{x}_i)$. Then,

$$\mathbb{E}_{u \sim [0,1]}[S(\boldsymbol{x}_i, y_i, u)] = \mathbb{E}_{u \sim [0,1]}[\tilde{S}(\boldsymbol{x}_i, y_i, u)]$$

Based on Theorem 3, $s_i$ and $\tilde{s}_i$ are uniform condition $\boldsymbol{x}_i$, and we can get that $\tau = \tilde{\tau} = 1 - \alpha$. Next, for a test point $\boldsymbol{x}$, considering two scenarios, i.e., $\pi_{(1)}(\boldsymbol{x}) \geq \tau$ and $\pi_{(1)}(\boldsymbol{x}) < \tau$. If $\pi_{(1)}(\boldsymbol{x}) \geq \tau$, the probability of $u \cdot \pi_{(1)}(\boldsymbol{x}) \leq \tau$ is $\frac{\tau}{\pi_{(1)}(\boldsymbol{x})}$. Thus,

$$\mathbb{E}_{u\sim[0,1]}\{|\tilde{\mathcal{C}}(\boldsymbol{x},u)|\} = \mathbb{E}_{u\sim[0,1]}\{|\mathcal{C}(\boldsymbol{x},u)|\} = \frac{\tau}{\pi_{(1)}(\boldsymbol{x})}$$

If $\pi_{(1)}(\boldsymbol{x}) < \tau$, without loss of generality, supposed $\sum_{k=1}^m \pi_{(k)}(\boldsymbol{x}) < \tau$ and $\sum_{k=1}^{m+1} \pi_{(k)}(\boldsymbol{x}) > \tau$.

$$\mathbb{E}_{u\sim[0,1]}\{|\tilde{\mathcal{C}}(\boldsymbol{x},u)|\} = m + \frac{\tau - \sum_{k=1}^m \pi_{(k)}(\boldsymbol{x})}{\pi_{(m+1)}(\boldsymbol{x})}.$$

But for SAPS, the analysis is more complex. Since $\pi_{(1)}(\boldsymbol{x}) > \frac{1}{K}$, $\pi_{(1)}(\boldsymbol{x}) + \lambda \geq 1 > \tau$

$$\mathbb{E}_{u\sim[0,1]}\{|\mathcal{C}(\boldsymbol{x},u)|\} = 1 + \frac{\tau - \pi_{(1)}(\boldsymbol{x})}{\lambda}.$$

When $m \geq 2$, with $\frac{1}{\lambda} \leq \frac{1}{1-\frac{1}{K}}$ and $\pi_{(1)}(\boldsymbol{x}) \in [\frac{1}{K}, 1]$ proposed in Lemma 1,

$$\mathbb{E}_{u\sim[0,1]}\{|\mathcal{C}(\boldsymbol{x},u)|\} = 1 + \frac{\tau - \pi_{(1)}(\boldsymbol{x})}{\lambda} \leq 1 + \frac{\tau - \frac{1}{K}}{1 - \frac{1}{K}} < 2 < \mathbb{E}_{u\sim[0,1]}\{|\tilde{\mathcal{C}}(\boldsymbol{x},u)|\}.$$

When $m = 1$, with $\lambda > \pi_{(2)}(\boldsymbol{x})$ in Lemma 1, we can get

$$1 + \frac{\tau - \pi_{(1)}(\boldsymbol{x})}{\lambda} < 1 + \frac{\tau - \pi_{(1)}(\boldsymbol{x})}{\pi_{(2)}(\boldsymbol{x})}$$

Finally, $\mathbb{E}_{u\sim[0,1]}\{|\mathcal{C}(\boldsymbol{x},u)|\} \leq \mathbb{E}_{u\sim[0,1]}\{|\tilde{\mathcal{C}}(\boldsymbol{x},u)|\}$. □

**Remark 1.** *Here, we discuss SAPS with a large $\lambda$. As the rank weight $\lambda$ increases, the information of rank dominates the score function Eq. 8, resulting in Eq. 8 being closing to APS without probability value. In other words, when $\lambda \to \infty$, the set size of SAPS converges to a constant. The corresponding results have been shown in Figure 4a. For example, On the ViT model, when $\log \lambda$ exceeds 2, the set size keeps a stable value and converges to a black marker which represents the result of APS without probability value.*

## D  DETAILED RESULTS FOR COVERAGE RATE AND SET SIZE

In this section, we report the detailed results of coverage rate and set size on different datasets in Table 3,4, and 5. The median-of-means for each result is reported over ten different trials. The average results of multiple models have been reported in the main paper.

Table 3: Results on Imagenet. The median-of-means for each column is reported over 10 different trials. **Bold** numbers indicate optimal performance.

|  | $\alpha = 0.1$ | | | | | | $\alpha = 0.05$ | | | | | |
|  | Coverage | | | Size ↓ | | | Coverage | | | Size ↓ | | |
| Datasets | APS | RAPS | SAPS | APS | RAPS | SAPS | APS | RAPS | SAPS | APS | RAPS | SAPS |
| ResNeXt101 | 0.899 | 0.902 | 0.901 | 19.49 | 2.01 | **1.82** | 0.950 | 0.951 | 0.950 | 46.58 | 4.24 | **3.83** |
| ResNet152 | 0.900 | 0.900 | 0.900 | 10.51 | 2.10 | **1.92** | 0.950 | 0.950 | 0.950 | 22.65 | 4.39 | **4.07** |
| ResNet101 | 0.898 | 0.900 | 0.900 | 10.83 | 2.24 | **2.07** | 0.948 | 0.949 | 0.950 | 23.20 | 4.78 | **4.34** |
| ResNet50 | 0.899 | 0.900 | 0.900 | 12.29 | 2.51 | **2.31** | 0.948 | 0.950 | 0.950 | 25.99 | 5.57 | **5.25** |
| ResNet18 | 0.899 | 0.900 | 0.900 | 16.10 | 4.43 | **4.00** | 0.949 | 0.950 | 0.950 | 32.89 | 11.75 | **10.47** |
| DenseNet161 | 0.900 | 0.900 | 0.900 | 12.03 | 2.27 | **2.08** | 0.949 | 0.950 | 0.951 | 28.06 | 5.11 | **4.61** |
| VGG16 | 0.897 | 0.901 | 0.900 | 14.00 | 3.59 | **3.25** | 0.948 | 0.950 | 0.949 | 27.55 | 8.80 | **7.84** |
| Inception | 0.900 | 0.902 | 0.902 | 87.93 | 5.32 | **4.58** | 0.949 | 0.951 | 0.950 | 167.98 | 18.71 | **14.43** |
| ShuffleNet | 0.900 | 0.899 | 0.900 | 31.77 | 5.04 | **4.54** | 0.949 | 0.950 | 0.950 | 69.39 | 16.13 | **14.05** |
| ViT | 0.900 | 0.898 | 0.900 | 10.55 | 1.70 | **1.61** | 0.950 | 0.949 | 0.950 | 31.75 | 3.91 | **3.21** |
| DeiT | 0.901 | 0.900 | 0.900 | 8.51 | 1.48 | **1.41** | 0.950 | 0.949 | 0.949 | 24.88 | 2.69 | **2.49** |
| CLIP | 0.899 | 0.900 | 0.900 | 17.45 | 6.81 | **6.23** | 0.951 | 0.949 | 0.949 | 35.09 | 16.79 | **16.07** |
| average | 0.899 | 0.900 | 0.900 | 20.95 | 3.29 | **2.98** | 0.949 | 0.950 | 0.950 | 44.67 | 8.57 | **7.55** |

Table 4: Results on CIFAR-100. The median-of-means for each column is reported over 10 different trials. **Bold** numbers indicate optimal performance.

| | $\alpha = 0.1$ | | | | | | $\alpha = 0.05$ | | | | | |
| | Coverage | | | Size ↓ | | | Coverage | | | Size ↓ | | |
| Datasets | APS | RAPS | SAPS | APS | RAPS | SAPS | APS | RAPS | SAPS | APS | RAPS | SAPS |
|---|---|---|---|---|---|---|---|---|---|---|---|---|
| ResNet18 | 0.898 | 0.901 | 0.898 | 10.03 | 2.72 | **2.41** | 0.949 | 0.950 | 0.951 | 16.76 | 5.88 | **4.96** |
| ResNet50 | 0.896 | 0.902 | 0.899 | 6.51 | 2.16 | **2.04** | 0.946 | 0.948 | 0.946 | 12.49 | 4.37 | **3.69** |
| ResNet101 | 0.899 | 0.901 | 0.898 | 6.52 | 2.10 | **1.99** | 0.951 | 0.947 | 0.948 | 12.26 | 4.49 | **3.85** |
| DenseNet161 | 0.898 | 0.898 | 0.897 | 8.07 | 2.02 | **1.77** | 0.948 | 0.949 | 0.950 | 14.35 | 3.61 | **3.33** |
| VGG16 | 0.900 | 0.896 | 0.895 | 5.80 | 4.58 | **3.64** | 0.951 | 0.949 | 0.948 | 11.83 | 11.32 | **9.27** |
| Inception | 0.902 | 0.902 | 0.902 | 12.01 | 2.01 | **2.01** | 0.952 | 0.953 | 0.953 | 18.24 | 5.21 | **4.26** |
| ViT | 0.897 | 0.901 | 0.900 | 4.29 | 2.14 | **1.91** | 0.949 | 0.948 | 0.949 | 7.92 | 3.98 | **3.57** |
| CLIP | 0.899 | 0.900 | 0.900 | 9.84 | 6.18 | **5.58** | 0.952 | 0.948 | 0.950 | 16.04 | 12.50 | **11.27** |
| average | 0.899 | 0.900 | 0.899 | 7.88 | 2.99 | **2.67** | 0.950 | 0.949 | 0.949 | 13.74 | 6.42 | **5.53** |

Table 5: Results on CIFAR-10. The median-of-means for each column is reported over 10 different trials. **Bold** numbers indicate optimal performance.

| | $\alpha = 0.1$ | | | | | | $\alpha = 0.05$ | | | | | |
| | Coverage | | | Size ↓ | | | Coverage | | | Size ↓ | | |
| Datasets | APS | RAPS | SAPS | APS | RAPS | SAPS | APS | RAPS | SAPS | APS | RAPS | SAPS |
|---|---|---|---|---|---|---|---|---|---|---|---|---|
| ResNet18 | 0.896 | 0.896 | 0.898 | 2.42 | 2.29 | **2.18** | 0.948 | 0.947 | 0.949 | 3.12 | 3.12 | **3.10** |
| ResNet50 | 0.897 | 0.900 | 0.895 | 2.08 | 1.95 | **1.77** | 0.949 | 0.950 | 0.950 | 2.69 | 2.62 | **2.53** |
| ResNet101 | 0.899 | 0.901 | 0.898 | 2.15 | 2.00 | **1.86** | 0.949 | 0.948 | 0.949 | 2.78 | 2.68 | **2.59** |
| DenseNet161 | 0.898 | 0.899 | 0.898 | 2.06 | 1.90 | **1.71** | 0.949 | 0.953 | 0.949 | 2.67 | 2.53 | **2.35** |
| VGG16 | 0.897 | 0.900 | 0.897 | 1.75 | 1.52 | **1.39** | 0.950 | 0.949 | 0.949 | 2.22 | 2.02 | **1.87** |
| Inception | 0.904 | 0.904 | 0.902 | 2.28 | 2.04 | **1.78** | 0.952 | 0.953 | 0.952 | 3.04 | 2.78 | **2.55** |
| ViT | 0.901 | 0.899 | 0.900 | 1.50 | 1.34 | **1.21** | 0.952 | 0.952 | 0.950 | 1.89 | 1.79 | **1.58** |
| CLIP | 0.897 | 0.900 | 0.899 | 1.51 | 1.27 | **1.16** | 0.949 | 0.949 | 0.950 | 1.89 | 1.57 | **1.41** |
| average | 0.899 | 0.900 | 0.898 | 1.97 | 1.79 | **1.63** | 0.950 | 0.950 | 0.950 | 2.54 | 2.39 | **2.25** |

## E  ESCV ON CIAFR-10 AND CIFAR-100

In this section, we report the results of ESCV on CIFAR-10 and CIFAR-100 with $\alpha = 0.1$. From Figure 5a and 5b, we can observe that the SAPS still can get smaller ESCV than RAPS. Moreover, as the scale of the classification task increases, the ESCV for RAPS also increases significantly, while SAPS maintains a low value. For example, on CLIP, the ESCV for RAPS on ImageNet is higher than that for CIFAR-100. But, for both ImageNet and CIFAR-100, the ESCV of SAPS remains around $0.35$.

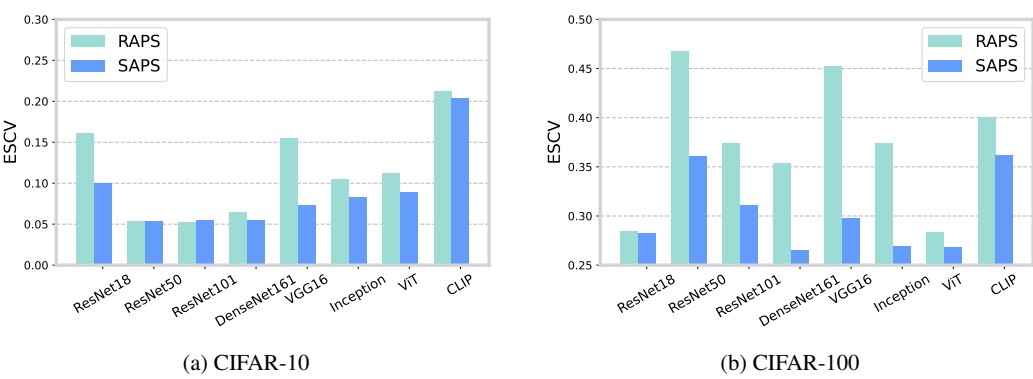

(a) CIFAR-10        (b) CIFAR-100

Figure 5: ESCV with different models on CIFAR-10 and CIFAR-100 with $\alpha = 0.1$.

# F ADAPTATION

In this section, we report detailed results of adaptation for four models with different architectures on ImageNet. In Table 6, for all models, SAPS achieves a smaller set size for easy examples than SAPS, but a larger set size for hard examples.

Table 6: Size conditional on difficulty. We report the size of RAPS and SAPS with ResNet152, VGG16, Inception, and ViT on ImageNet. The term "Difficulty" refers to the relative ranking of the probability estimates of the true label class. The "Count" corresponds to the number of examples falling within each difficulty category.

| | ResNet152 | | | VGG16 | | | Inception | | | ViT | | |
|---|---|---|---|---|---|---|---|---|---|---|---|---|
| | Count | Size | | Count | Size | | Count | Size | | Count | Size | |
| Difficulty | | RAPS | SAPS | | RAPS | SAPS | | RAPS | SAPS | | RAPS | SAPS |
| 1 | 15696 | 1.87 | 1.57 | 14316 | 3.26 | 2.57 | 13905 | 4.72 | 3.32 | 16207 | 1.56 | 1.36 |
| 2-3 | 2560 | 2.79 | 2.85 | 3019 | 4.63 | 4.82 | 3058 | 6.63 | 6.81 | 2404 | 2.55 | 2.52 |
| 4-6 | 712 | 3.06 | 3.29 | 1013 | 5.11 | 5.49 | 997 | 7.36 | 8.19 | 598 | 2.78 | 2.78 |
| 7-10 | 345 | 3.10 | 3.39 | 455 | 5.23 | 5.72 | 584 | 7.47 | 8.57 | 267 | 2.81 | 2.96 |
| 11-100 | 615 | 3.19 | 3.51 | 1057 | 5.34 | 6.01 | 1116 | 7.77 | 9.09 | 452 | 3.05 | 3.26 |
| 101-1000 | 72 | 3.50 | 3.83 | 140 | 5.50 | 6.12 | 340 | 7.51 | 8.85 | 72 | 2.76 | 2.72 |

# G SAPS vs. RAPS($k_r = 1$)

To further understand the influence of excluding the probability value, we conduct an experiment for RAPS($k_r = 1$), which is similar to SAPS on ImageNet, CIFAR-100, and CIFAR-10 with $\alpha = 0.1$. The median results on ten trials are reported in Table 7, 8 and 9.

Table 7: Comparsion between RAPS($k_r = 1$) and SAPS on ImageNet with $\alpha = 0.1$. The median-of-means for each column is reported over 10 different trials. **Bold** numbers indicate optimal performance.

| | $\alpha = 0.1$ | | | | | | $\alpha = 0.05$ | | | | | |
|---|---|---|---|---|---|---|---|---|---|---|---|---|
| | Coverage | | Size ↓ | | ESCV ↓ | | Coverage | | Size ↓ | | ESCV ↓ | |
| Datasets | RAPS ($k_r = 1$) | SAPS | RAPS ($k_r = 1$) | SAPS | RAPS ($k_r = 1$) | SAPS | RAPS ($k_r = 1$) | SAPS | RAPS ($k_r = 1$) | SAPS | RAPS ($k_r = 1$) | SAPS |
| ResNeXt101 | 0.902 | 0.901 | 1.85 | **1.82** | 0.477 | **0.366** | 0.950 | 0.950 | 4.56 | **3.83** | 0.505 | **0.358** |
| ResNet152 | 0.901 | 0.900 | 1.97 | **1.92** | 0.572 | **0.440** | 0.950 | 0.950 | 4.68 | **4.07** | 0.750 | **0.335** |
| ResNet101 | 0.900 | 0.900 | 2.09 | **2.07** | 0.650 | **0.431** | 0.950 | 0.950 | 4.95 | **4.34** | 0.724 | **0.466** |
| ResNet50 | 0.900 | 0.900 | 2.32 | **2.31** | 0.594 | **0.350** | 0.950 | 0.950 | 5.52 | 5.25 | 0.950 | **0.373** |
| ResNet18 | 0.899 | 0.900 | 4.50 | **4.00** | 0.900 | **0.409** | 0.950 | 0.950 | **10.03** | 10.47 | 0.950 | **0.490** |
| DenseNet161 | 0.901 | 0.900 | 2.12 | **2.08** | 0.531 | **0.393** | 0.951 | 0.951 | 5.15 | **4.61** | 0.950 | **0.405** |
| VGG16 | 0.899 | 0.900 | 3.69 | **3.25** | 0.671 | **0.425** | 0.949 | 0.949 | **7.56** | 7.84 | 0.950 | **0.390** |
| Inception | 0.901 | 0.902 | 5.13 | **4.58** | 0.478 | **0.418** | 0.951 | 0.950 | **13.20** | 14.43 | 0.950 | **0.532** |
| ShuffleNet | 0.900 | 0.900 | 5.11 | **4.54** | 0.481 | **0.375** | 0.951 | 0.950 | **12.38** | 14.05 | 0.473 | **0.306** |
| ViT | 0.899 | 0.900 | **1.58** | 1.61 | 0.900 | **0.393** | 0.950 | 0.950 | 3.32 | **3.21** | 0.950 | 0.950 |
| DeiT | 0.899 | 0.900 | **1.38** | 1.41 | 0.416 | **0.389** | 0.950 | 0.949 | 2.59 | **2.49** | 0.596 | **0.418** |
| CLIP | 0.900 | 0.900 | 7.16 | **6.23** | 0.900 | **0.369** | 0.949 | 0.949 | **15.16** | 16.07 | 0.950 | **0.317** |
| average | 0.900 | 0.900 | 3.24 | **2.98** | 0.631 | **0.396** | 0.950 | 0.950 | **7.43** | 7.55 | 0.808 | **0.445** |

Table 8: Comparsion between RAPS($k_r = 1$) and SAPS on CIFAR-100 with $\alpha = 0.1$. The median-of-means for each column is reported over 10 different trials. **Bold** numbers indicate optimal performance.

| | $\alpha = 0.1$ | | | | | | $\alpha = 0.05$ | | | | | |
| --- | --- | --- | --- | --- | --- | --- | --- | --- | --- | --- | --- | --- |
| | Coverage | | Size ↓ | | ESCV ↓ | | Coverage | | Size ↓ | | ESCV ↓ | |
| Datasets | RAPS ($k_r = 1$) | SAPS | RAPS ($k_r = 1$) | SAPS | RAPS ($k_r = 1$) | SAPS | RAPS ($k_r = 1$) | SAPS | RAPS ($k_r = 1$) | SAPS | RAPS ($k_r = 1$) | SAPS |
| ResNet18 | 0.897 | 0.898 | 2.51 | **2.41** | 0.321 | **0.282** | 0.951 | 0.951 | 6.30 | **4.96** | **0.209** | 0.244 |
| ResNet50 | 0.900 | 0.899 | **1.97** | 2.04 | 0.383 | **0.360** | 0.948 | 0.946 | 4.52 | **3.69** | 0.396 | **0.200** |
| ResNet101 | 0.899 | 0.898 | **1.92** | 1.99 | 0.344 | **0.311** | 0.950 | 0.948 | 4.56 | **3.85** | 0.210 | **0.242** |
| DenseNet161 | 0.899 | 0.897 | **1.75** | 1.77 | 0.381 | **0.265** | 0.949 | 0.950 | 3.69 | **3.33** | 0.303 | **0.298** |
| VGG16 | 0.897 | 0.895 | 4.06 | **3.64** | 0.567 | **0.298** | 0.950 | 0.948 | 10.75 | **9.27** | 0.450 | **0.325** |
| Inception | 0.901 | 0.902 | **1.86** | 2.01 | 0.375 | **0.269** | 0.953 | 0.953 | 4.57 | **4.26** | 0.305 | **0.213** |
| ViT | 0.902 | 0.900 | **1.89** | 1.91 | 0.312 | **0.268** | 0.946 | 0.949 | 4.01 | **3.57** | 0.190 | **0.170** |
| CLIP | 0.900 | 0.900 | 6.38 | **5.58** | 0.438 | **0.362** | 0.949 | 0.950 | **10.47** | 11.27 | **0.406** | 0.429 |
| average | 0.899 | 0.899 | 2.79 | **2.67** | 0.390 | **0.302** | 0.949 | 0.949 | 6.11 | **5.53** | 0.309 | **0.265** |

Table 9: Comparsion between RAPS($k_r = 1$) and SAPS on CIFAR-10 with $\alpha = 0.1$. The median-of-means for each column is reported over 10 different trials. **Bold** numbers indicate optimal performance.

| | $\alpha = 0.1$ | | | | | | $\alpha = 0.05$ | | | | | |
| --- | --- | --- | --- | --- | --- | --- | --- | --- | --- | --- | --- | --- |
| | Coverage | | Size ↓ | | ESCV ↓ | | Coverage | | Size ↓ | | ESCV ↓ | |
| Datasets | RAPS ($k_r = 1$) | SAPS | RAPS ($k_r = 1$) | SAPS | RAPS ($k_r = 1$) | SAPS | RAPS ($k_r = 1$) | SAPS | RAPS ($k_r = 1$) | SAPS | RAPS ($k_r = 1$) | SAPS |
| ResNet18 | 0.897 | 0.898 | 2.24 | **2.18** | 0.145 | **0.100** | 0.946 | 0.949 | **2.98** | 3.10 | **0.052** | 0.076 |
| ResNet50 | 0.899 | 0.895 | 1.79 | **1.77** | 0.101 | **0.054** | 0.949 | 0.950 | 2.63 | **2.53** | 0.062 | **0.030** |
| ResNet101 | 0.899 | 0.898 | 1.87 | **1.86** | 0.107 | **0.054** | 0.946 | 0.949 | 2.61 | **2.59** | 0.043 | **0.026** |
| DenseNet161 | 0.899 | 0.898 | **1.69** | 1.71 | 0.156 | **0.055** | 0.952 | 0.949 | 2.52 | **2.35** | 0.103 | **0.021** |
| VGG16 | 0.900 | 0.897 | **1.38** | 1.39 | 0.112 | **0.073** | 0.949 | 0.949 | **1.84** | 1.87 | 0.124 | **0.050** |
| Inception | 0.903 | 0.902 | **1.77** | 1.78 | 0.143 | **0.083** | 0.954 | 0.952 | 2.68 | **2.55** | 0.089 | **0.040** |
| ViT | 0.900 | 0.900 | **1.16** | 1.21 | 0.127 | **0.089** | 0.950 | 0.950 | 1.63 | **1.58** | 0.101 | **0.054** |
| CLIP | 0.899 | 0.899 | **1.06** | 1.16 | 0.213 | **0.203** | 0.950 | 0.950 | **1.36** | 1.41 | 0.254 | **0.123** |
| average | 0.900 | 0.898 | **1.62** | 1.63 | 0.138 | **0.089** | 0.949 | 0.950 | 2.28 | **2.25** | 0.104 | **0.052** |

# H PROCEDURE OF SAPS

This section describes the whole procedure of SAPS. The prediction sets are defined as

$$\mathcal{C}_{1-\alpha}(\boldsymbol{x}, u) := \{y \in \mathcal{Y} : S(\boldsymbol{x}, y, u; \hat{\pi}) \leq \tau\}, \tag{12}$$

where $S(\boldsymbol{x}, y, u; \hat{\pi})$ is the score function defined as in Eq. 8. Then, we provide the main procedure by Algorithm 1, the selection procedure of $\lambda$ by Algorithm 2, and the calibration procedure by Algorithm 3. In this work, we conduct RAPS in a similar procedure.

---

**Algorithm 1** SAPS

---

**Input:** Error rate $\alpha$, classifier $\hat{\pi}$, a calibration set $\mathcal{D}_{cal}$, a test point $\boldsymbol{x}$;
**Output:** a prediction set $\mathcal{C}$;
  1: Randomly sample a validation subset $\mathcal{D}_{val}$ from the dataset $\mathcal{D}_{cal}$;
  2: $\lambda \leftarrow$ the output of Algorithm 2 with inputs $(\alpha, \hat{\pi}, \mathcal{D}_{val})$;
  3: $\tau \leftarrow$ the output of Algorithm 3 with inputs $(\alpha, \hat{\pi}, \mathcal{D}_{cal} \setminus \mathcal{D}_{val}, \lambda)$;
  4: $u \leftarrow U[0, 1]$;
  5: Generate the prediction set $\mathcal{C}$ for $\boldsymbol{x}$ by Eq. 12.

---

---

**Algorithm 2** SAPS Ranking Weight

---

**Input:** Error rate $\alpha$, classifier $\hat{\pi}$, a data set $\mathcal{D}$
**Output:** a real value $\lambda^*$;
1: Configure a parameters set $\mathcal{D}_\lambda$;
2: **for all** $i \in \{1, 2, \ldots, |\mathcal{D}_\lambda|\}$ **do**
3:     Calculate $\tau$ by Algorithm 3 with inputs $(\alpha, \hat{\pi}, \mathcal{D}, \lambda_i)$;
4:     For each $j \in \mathcal{D}$ and $u_j \sim U[0,1]$, calculate its prediction set $\mathcal{C}(\boldsymbol{x}_j, u_j)$ by Eq. 12;
5:     $E_i \leftarrow$ the mean size of sets $\{\mathcal{C}(\boldsymbol{x}_j, u_j)\}_{j \in \mathcal{D}}$ ;
6: **end for**
7: $\lambda^* \leftarrow$ the $\lambda_i$ whose mean size is the smallest value of $\{E_i\}_{i \in \mathcal{D}_\lambda}$.

---

**Algorithm 3** SAPS Conformal Calibration

---

**Input:** Error rate $\alpha$, classifier $\hat{\pi}$, a data set $\mathcal{D}$, a ranking weight $\lambda$;
**Output:** the calibrated threshold $\tau$;
    $n \leftarrow |\mathcal{D}|$;
    **for all** $i = 1, 2, \cdots, n$ **do**
        $u_i \sim U[0,1]$;
        $s_i = S(\boldsymbol{x}_i, y_i, u_i; \hat{\pi})$; // The score function is defined as in Eq. 8
    **end for**
    $\tau \leftarrow$ the $\lceil (1-\alpha)(1+n) \rceil$ largest value in $\{s_i\}_{i=1}^n$.

---

