# OpenReview forum: "Conformal Prediction for Deep Classifier via Label Ranking"
_ICLR.cc/2024/Conference — Submitted to ICLR 2024_

### Official Review · Reviewer_SVnn · 2023-10-26

**Soundness:** 3 good
**Presentation:** 4 excellent
**Contribution:** 3 good
**Rating:** 8
**Confidence:** 4

**Summary:**

This paper mainly proposes a new nonconformity score for classification task. It first identifies that the probability in APS is not that important and the rank instead is important - a rank-based version of the APS nonconformity score makes more efficient prediction sets, the size of which is also more correlated with accuracy. Then, it proposes a new nonconformity score that only keeps the top predicted probability but uses ranks for the remaining classes, and shows in the experiments that such score typically improves efficiency.

**Strengths:**

This is a well-motivated paper that is very easy to follow. The observation that ranking matters more than the predicted probability (or softmax output) is very interesting (and I'm surprised that it has not been discovered before). The proposed solution also makes sense. Proposition 2 about dominance is a good theoretical result on top of the finite-sample coverage.

**Weaknesses:**

This requires choosing a hyperparameter $\lambda$, which requires additional data and could affect efficiency in practice.

**Questions:**

In the experiments, choosing $\lambda$ for SAPS uses a subset of the calibration set. Do all baselines uses the (same) remaining calibration set? That is, is SAPS calibrated on a smaller set due to $\lambda$?

---

> ### Author Response · Authors · 2023-11-19
> **Response to Reviewer SVnn**
>
> Thank you for the recognition. We address specific concerns below.
>
>
> **1. Choice of $\lambda$.  [Q1]**
>
> For SAPS and RAPS, we utilize a validation set to tune hyperparameters and the remaining dataset to calibrate the $\tau$. For APS, we calibrate the threshold $\tau$ on the whole calibration set.

---

### Official Review · Reviewer_poFk · 2023-10-27

**Soundness:** 3 good
**Presentation:** 2 fair
**Contribution:** 2 fair
**Rating:** 5
**Confidence:** 3

**Summary:**

Using all softmax probabilities in the non-conformity score may yield larger prediction sets. Out of this consideration, the authors proposed a method called sorted adaptive prediction sets (SAPS), which discards all the probability values except the maximum softmax probability.

**Strengths:**

* The authors empirically showed that using all probabilities is not necessary in APS.
* The authors empirically showed that, under different network architectures, the proposed method returns more efficient prediction sets compared to APS and RAPS on three datasets.

**Weaknesses:**

* I have a reservation about the claimed contribution of higher adaption, i.e., the adaption is not that convincing: For the example of Figure 3(b), now that both SAPS and RAPS achieve the same coverage, why should we require a larger prediction set for difficult observations? In general, the smaller the better. RAPS gives more efficient predictions on those observations with higher difficulty, while the proposed SAPS only gives efficient predictions on a shorter interval of difficulty.

* Even though the proposed method has a promising performance compared to several methods, how is the proposed method far away from the ground truth?

* To make it clearly catch the whole scope, it would be better to explicitly outline the calibration and prediction under the frame of a pseudo-algorithm as the one in RAPS. For example, I believe "We choose the hyper-parameter that achieves the smallest set size on a validation set" fails to disclose the entire picture because the smallest set size on a validation set cannot secure the desired coverage. How did the authors handle this issue?

* The proofs are not friendly reading (see the section on questions).

* Minor issues:
  * Do you mean $\mathcal{C}(\boldsymbol x_i, y_i, u_i)$ for the definition of coverage rate?
  * Double-check all the usage of "i.e.,"
  * 0.05 instead of 0.5 in Section 4.2.
  * In the proof of Lemma 1, did you intend to assume $p_{(k)}\geq \frac{1}{k}$? Where will be $\tilde{k}$ used in the proof?

**Questions:**

* Is (2) generally correct? In other words, are the prediction results always nested? Particularly in Theorem 2, Since there is a random variable $u$ introduced, why $\mathcal{C}_{1-\alpha}(\boldsymbol{x}, u)$ have the nesting property?
* Proposition 1: How is $\mathcal{C}_{1-\alpha}(\boldsymbol x, u)$ defined as in Eq. 3? They have totally different notations.
* I didn’t get the point of the proof for Proposition 1. What is the difference between your proof of proposition 1 and Theorem 2? The conclusion of coverage is for the popped $\mathcal{C}(\boldsymbol x_{n+1},u_{n+1})$ but there is no $\mathcal{C}(\boldsymbol x_{n+1},u_{n+1})$ during your proof. I think the authors need to well-articulate the proof.
* Why $\frac{1}{\lambda}\leq1-\frac{1}{K}$ but previously you require $\lambda>1-\frac{1}{K}$?

**Details Of Ethics Concerns:**

No.

---

> ### Author Response · Authors · 2023-11-19
> **Response to Reviewer poFk**
>
> Thanks for your recognition and the valuable suggestions. We give a detailed response to your questions and comments in the following.
>
>
> **1. Adaption of sets.[W1]**
>
> In the literature of conformal prediction, the size of prediction sets is expected to represent the inherent uncertainty of the classifier's predictions. With a comparable set size, methods with high adaption can reflect instance-wise uncertainty precisely [R1]. Specifically, prediction sets should be larger for hard examples than for easy ones. Adaption is particularly significant in high-stakes scenarios where assessing the model's reliability is necessary. In Figure 3b, we show SAPS preserves more distinguishable information than RAPS, while achieving a smaller average set size and desired coverage rate. This advantage further strengthens the value of the proposed method.
>
>
>
> **2. Gap between SAPS and ground truth [W2]**
>
> We guess that the "ground truth" is the sets produced by APS with the oracle model (correct us if we are mistaken). In Proposition 2, we provide a theoretical result, suggesting that SAPS is capable of producing smaller sets than APS, with the oracle model.
>
>
> **3. The desired coverage [W3]**
>
> Thank you for the great suggestion. We have included pseudo-code algorithms in Appendix H of the revised manuscript.
>
> For the validation set, we would like to clarify that various score functions caused by different values of $\lambda$ always satisfy the desired coverage. In other words, the value of $\lambda$ does not affect the coverage guarantee, which is theoretically proved by Proposition 1. This is also supported by the empirical results in Sec 4.2.
>
>
> **4. Minor issues [W4,W5,Q3,Q4]**
>
> We thanks for pointing out the typos. We have fixed them in the updated version.
> - $\mathcal{C}\left(\boldsymbol{x}\_{i},y_i,u_i\right)$ is a typo. In Section 4.1, we revised $\mathcal{C}\left(\boldsymbol{x}\_{i},y_i,u_i\right)$ to $\mathcal{C}\left(\boldsymbol{x}\_{i}\right)$ representing the  prediction set for $\boldsymbol{x}\_{i}$.
> - In the proof of Lemma 1, we assume that $p_{(k)}>\frac{1}{k}$ for any $k\geq 2$. Thus,  for any $\tilde{k}< k$, it holds that $p_{(\tilde{k})}>\frac{1}{k}$.
> - In Proposition 1, the subscript of symbol $\mathcal{C}\_{1-\alpha}(\boldsymbol{x}\_{n+1},u\_{n+1})$  hase been completed.
> - In proof of Proposition 2, the inequality is the $\frac{1}{\lambda} \leq \frac{1}{1-\frac{1}{K}}$.
>
> **5. Definition of prediction set in Proposition 1 [Q2]**
>
> Thank you for pointing out the ambiguous notation. The prediction set in Proposition 1 is defined as  $$\mathcal{C}_{1-\alpha}(\boldsymbol{x},u) =\lbrace y\in\mathcal{Y} : S(\boldsymbol{x},y,u;\hat{\pi})\leq \tau \rbrace.$$ Thus, it is mathematically equivalent to Eq. 3. To mitigate this confusion, we added a detailed description in Proposition 1.
>
> **6. The nest property [Q1]**
>
> The nesting property defined by Eq.2 is a common property holding on all prediction sets for any conformal predictor [R2]. Specifically, if a lower error rate is expected, the set will have a larger size for higher coverage. In this work, since the calibrated threshold $\tau$ is the $1-\alpha$ quantile of scores, the nesting property of $\alpha$ is equivalent to a nesting property for $\tau$, i.e., $$\alpha_1>\alpha_2 \rightarrow \tau_1\leq\tau_2 \rightarrow \lbrace y\in\mathcal{Y}: S(\boldsymbol{x},y,u;\hat{\pi})\leq \tau_1 \rbrace \subseteq \lbrace y\in\mathcal{Y}: S(\boldsymbol{x},y,u;\hat{\pi})\leq \tau_2\rbrace,$$ for a test point $\boldsymbol{x}$ and a random variable $u$. Therefore, we get the nesting property of $\mathcal{C}_{1-\alpha}(\boldsymbol{x},u)$.
>
>
> **7. Difference between proposition 1 and Theorem 2  [Q3]**
>
> Proposition 1 is a corollary of Theorem 2. Specifically, Theorem 2 gives a coverage guarantee for CP methods whose prediction set has a general formulation $\mathcal{C}(\boldsymbol{x},u,\tau)$. The prediction set of SAPS defined as $\lbrace y\in\mathcal{Y}:S(\boldsymbol{x},y,u;\hat{\pi})\leq \tau \rbrace$ is a specific instance of $\mathcal{C}(\boldsymbol{x},u,\tau)$. Therefore, Proposition 1 offers a coverage guarantee for SAPS. We apologize for this misunderstanding and have revised  Appendix B to make this clearer.
>
> [R1] Anastasios Nikolas Angelopoulos, Stephen Bates, Michael I. Jordan, and Jitendra Malik. Uncertainty sets for image classifiers using conformal prediction. In 9th International Conference on Learning Representations
>
> [R2] Balasubramanian, V., Ho, S.-S., and Vovk, V. (2014). Conformal prediction for reliable machine learning: theory, adaptations and applications.

---

> ### Author Response · Authors · 2023-11-22
> **Sincerely expect your further comments**
>
> Dear reviewer poFK,
>
> Sorry to disturb you. We sincerely appreciate your valuable comments. We would like to further provide brief answers here to the issues that might be your primary concerns.
>
> We really appreciate your efforts in identifying the typos and ambiguous descriptions, as well as providing valuable suggestions for method descriptions. In response to your feedback, we have carefully revised the typos and enhanced the clarity of our descriptions. We have provided a specific description of the technique details of our proposed method in Appendix H.
>
> We guess that you may have concerns about adaption. With a comparable set size, methods with high adaption can discriminately reflect instance-wise uncertainty. In particular, the results show that our method preserves more distinguishable information than RAPS, while achieving a smaller average set size and desired coverage rate.
>
> In addition, we would like to clarify that various score functions caused by different values of $\lambda$ always satisfy the desired coverage. This concern can be resolved by Proposition 1 and verified by empirical results.
>
> We sincerely hope that the above answers can address your concerns. We look forward to your response and are willing to answer any questions.
>
> Thank you

---

### Official Review · Reviewer_xE2a · 2023-10-30

**Soundness:** 3 good
**Presentation:** 3 good
**Contribution:** 3 good
**Rating:** 5
**Confidence:** 3

**Summary:**

Conformal prediction is a widely used framework, which can outputs a confidence set with statistical guarantee. A crucial aspect of this framework is the choice of non-conformity measure. The authors modify the Adaptive Prediction Set (APS) and propose a novel non-conformity measure called Sorted Adaptive Prediction Set (SAPS). The authors theoretically demonstrate that this non-conformity measure maintains finite-sample marginal coverage and dominates APS in terms of prediction set size in some special cases. Empirically, they show the superiority of the proposed method over APS and RAPS across different datasets

**Strengths:**

1. The authors propose a novel non-conformity measure in classification problem, and theoretically show it always dominates APS in the size of prediction sets if $\hat{\pi} = \pi$.
2. The authors conduct the experiments on three different datasets. They propose a novel metric ESCV to evaluate the performance of methods.

**Weaknesses:**

1. The theoretical contribution of this paper seems limited. Proposition 1 represents a common property of any non-conformity measure. Moreover, the condition in Proposition 2, $\hat{\pi} = \pi$, is challenging to satisfy in practice. As for another condition  $\lambda \geq 1 - \frac{1}{K}$, note that $\lambda$ used in experiments is searched in the range of 0.001 to 0.5, which conflicts with this condition. Figure 4a shows that when $\lambda$ exceeds 0.2, the set size increases with $\lambda$. It is important to address these concerns.
2. In Equation (8), it is unclear why the authors use 2 instead of another constant in $o(y,\hat{\pi}(x)) - 2 + u$. The authors should provide a motivation and explanation for this choice.
3. The authors only provide theoretical analysis comparing APS and SAPS, and empirical comparisons between RAPS and SAPS. It is necessary to include a detailed comparison with RAPS theoretically, since it is also a modified version of APS.

**Questions:**

Please see the Weaknesses.

---

> ### Author Response · Authors · 2023-11-19
> **Response to Reviewer xE2a (1/2)**
>
> Thank you for the valuable comments and detailed feedback on our manuscript. Please find our response below.
>
> **1. Proposition 1 represents a common property [W1]**
>
> Yes, many existing works have shown their methods possess the property [R1, R2] while some works do not provide theoretical proofs [R3, R4]. In this work, we present the proposition to show that the proposed method also satisfies the finite-sample coverage guarantee from a theoretical perspective. Without the proposition, it becomes ambiguous if our method can obtain the theoretical property. In other words, Proposition 1 enhances the completeness of our work.
>
>
> **2. Calibrated condition of Proposition 2**
>
> In the literature of conformal prediction, it is a common assumption that the given model is well-calibrated in theoretical analysis [R5, R6, R7]. Based on the assumption, we can then provide the subsequent analysis to show the inherent advantages of SAPS compared to APS. On the contrary, it would be non-trivial to give an in-depth understanding of the proposed method, without the condition. Thus, the theoretical explanation is still valuable to the community. In addition, the empirical results in Section 4 demonstrate the effectiveness of our method with imperfect models.
>
>
> **3. The range of $\lambda$ in Proposition 2**
>
> We would like to clarify that the range of 0.001 to 0.5 is for the hyperparameter of RAPS, instead of our method. We tune the $\lambda$ of SAPS in the range of 0.02 to 0.6. Moreover, the threshold of $\lambda$ in Proposition 2 is a sufficient condition to show the advantage of SAPS, but it does not reflect that the optimal value of $\lambda$ must exist in the interval $[1-\frac{1}{K},\infty)$ in the experiments.
>
> Here, we conduct an experiment to validate the effectiveness of SAPS with a $\lambda$ in the interval $[1-\frac{1}{K},\infty)$. In particular, we set $\lambda = 1$. The results on three datasets are shown in the table below. We show that SAPS with $\lambda=1$ still outperforms APS, which validates the proposition.
>
> |  Method   | APS  |  SAPS($\lambda=1$) |
> |  :----:  | :----:  | :----:  |
> | ImageNet  | 20.95 | 3.82     |
> | CIFAR-100  | 7.88 | 3.35    |
> | CIFAR-10  | 1.97 | 1.80   |
>
> Finally, we would like to clarify that the set size of SAPS would not maintain the upward trend with the increase of $\lambda$. Instead, the set size will converge to a specific value, and SAPS is equivalent to the APS without probability value when $\lambda \to \infty$. To demonstrate this, we update Figure 4a in the revised version by extending the value of $\lambda$ and the updated figure validates the above statement.
>
>
> **4. Clarification of proposed score function [W2]**
>
> The constant $2$ in Eq. (8) is not a user-guided value. The term $(o(y,\hat{\pi}(\boldsymbol{x}))-2+u)$ contains ranking scores from rank $2$ to rank $(o(y,\hat{\pi}(\boldsymbol{x}))-1)$, and a random ranking score at the rank $o(y,\hat{\pi}(\boldsymbol{x}))$. As the ranking score is defined as a constant $\lambda$, the sum of ranking scores from rank $2$ to rank $(o(y,\hat{\pi}(\boldsymbol{x}))-1)$ is $(o(y,\hat{\pi}(\boldsymbol{x}))-2) \cdot \lambda$. Thus, the formulation of the proposed score function is defined as $\hat{\pi}_{max} (\boldsymbol{x}) + (o(y,\hat{\pi}(\boldsymbol{x}))-2) \cdot \lambda$+ $u\cdot \lambda$.
>
> [R1] Anastasios Nikolas Angelopoulos, et al. Uncertainty sets for image classifiers using conformal prediction. In 9th International Conference on Learning Representations, ICLR 2021
>
> [R2] Charles Lu, et al. Federated conformal predictors for distributed uncertainty quantification. International Conference on Machine Learning, ICML 2023
>
> [R3] Subhankar Ghosh, et al. Improving uncertainty quantification of deep classifiers via neighborhood conformal prediction: Novel algorithm and theoretical analysis. AAAI 2023
>
> [R4] Henrik Linusson, et al. Classification with reject option using conformal prediction. PAKDD 2018
>
> [R5] Bat-Sheva Einbinder, et al. Training uncertainty-aware classifiers with conformalized deep learning. Advances in Neural Information Processing Systems, 2022b
>
> [R6] Aleksandr Podkopaev, et al. Distribution-free uncertainty quantification for classification under label shift. In Uncertainty in Artificial Intelligence
>
> [R7] Bat-Sheva Einbinder, et al. Conformal prediction is robust to label noise. arXiv preprint arXiv:2209.14295, 2022a

---

> ### Author Response · Authors · 2023-11-19
> **Response to Reviewer xE2a (2/2)**
>
> **5. Theoretical comparison with RAPS [W3]**
>
> In this work, we aim to provide an in-depth understanding of SAPS through the comparison of APS and RAPS. While there might be some new insights in the comparison between SAPS and RAPS, it is challenging to analyze these two methods in a unified framework, as RAPS introduces two hyper-parameters. Instead, we empirically show the superiority of SAPS over RAPS in set size, conditional coverage violation, and adaptation. Furthermore, we compare SAPS with RAPS($k_r=1$) in Sec 5 to further demonstrate the negative effect of probability values. We believe the thorough analysis of RAPS is sufficient to support our conclusion.

---

> ### Author Response · Authors · 2023-11-22
> **Sincerely expect your further comments**
>
> Dear Reviewer xE2a,
>
> Sorry to disturb you. We appreciate that you have pointed out some concerns. We believe that we have addressed all your concerns and clarified the misunderstanding part. Would you please kindly check that and consider re-evaluating our work? Please let us know if you have any further concerns and we are open to all possible discussions.
>
> Thank you

---

### Official Review · Reviewer_JXhd · 2023-10-30

**Soundness:** 4 excellent
**Presentation:** 4 excellent
**Contribution:** 4 excellent
**Rating:** 8
**Confidence:** 4

**Summary:**

This paper studies conformal prediction as applied to classification problems. It shows that one can significantly reduce the size of the set-valued prediction by removing the miscalibrated probability values in the long tail. This is done by discarding all the probability values except for the maximum softmax probability.

**Strengths:**

The paper is easy to read. The idea and the solution are both clearly articulated and the results are convincing. There are theoretical justifications.

**Weaknesses:**

See questions.

**Questions:**

1. On page 3, just above (4), it stated "In the APS method (Romano et al., 2019)". However, are you sure it's not Romano et al., 2020?  The 2019 paper was Conformalized quantile regression.

2. Page 13, right above equation (9), it stated "i.e., the calibrated threshold $\tau$, can be obtained by (an equation)". It is not obvious to me how this result was obtained and some clarification is appreciated. Moreover, what is the asterisk (*) at the end of that equation?

---

> ### Author Response · Authors · 2023-11-19
> **Response to Reviewer JXhd**
>
> Thank you for the recognition and valuable comments. Please find our response below.
> 1. **Correction of citation [Q1]**
>
> Thank you for the correction. We have fixed the citation in the revised version.
>
> 2. **Clarification  of the calibrated threshold [Q2]**
>
> Thank you for pointing out the unclear part. Here, we provide a detailed explanation of $\tau$. The score function is defined as $S(\boldsymbol{x},y,u;\hat{\pi}) =  r -u$ where $r$ is the rank of $\hat{\pi}\_y(\boldsymbol{x})$ in the sorted vector of $\hat{\pi}(\boldsymbol{x})$. We use $A_{r}$ to denote the proportion of examples in the calibration set that have scores less than $r$. If $k$ satisfies $A_{k}\geq 1-\alpha >A_{k-1}$, the $1-\alpha$ quantile of scores for the calibration set, i.e., the calibrated threshold $\tau$, falls within the interval $[k-1,k]$. As the scores in the interval $[k-1,k]$ are uniformly distributed, and the proportion of examples with scores in $[k-1,k]$ is $a_k$, $\tau$ is chosen as the $\frac{1-\alpha- A_{k-1}}{a_{k}}$ quantile of $[k-1,k]$. In the revised manuscript, we provide a detailed supplement for the proof in Appendix A. The asterisk (*) is a typo, which is fixed in the revised version.

---

### Author Response · Authors · 2023-11-19
**Gerenal Response**

We thank all the reviewers for their time, insightful suggestions, and valuable comments. We are glad that All reviewers appreciate that our method is **novel** and effective with **convincing results** on various datasets. We are also encouraged that reviewers appreciate the **well-motivation** of our work and its **coherence** (SVnn, JXhd). Additionally, Reviewers also regard our proposed method as having **theoretical justifications** (SVnn, xE2a, JXhd), bringing **interesting insight** to the community (SVnn), and introducing a new performance metric (xE2a).

In this rebuttal, we have given careful thought to the reviewers’ suggestions and made the following revisions to our manuscript to answer the questions and concerns:
- In Sections 2 and 3, we added the specific definition of prediction sets for various  methods and  further clarified the proposed propositions;
- In Section 5, We modified Figure 4a on more various $\lambda$;
- In Appendices A, B, and C, we clarified the missing details and typos;
- In Appendix H, we added pseudo-code algorithms about the implementation details.

We have highlighted the revised part in our manuscript in blue color. Please check the answers to specific comments.

---

### Meta-Review · Area_Chair_zWKx · 2023-12-09

**Metareview:**

Although the reviewers raise a number of critical points in their reports, there is agreement that the paper holds promise, and the authors' idea of a ranking-based approach to conformal prediction looks quite intriguing. The authors also showed a high level of commitment during the rebuttal phase and did their best to respond to the comments and to improve the submission. This was appreciated and positively acknowledged by all. In the discussion between authors and reviewers, some critical points could be resolved and some questions clarified. Other points remained open and were critically reconsidered in the subsequent internal discussion. In particular, some reviewers still find the theoretical contributions of the paper somewhat limited and question the practical usefulness of the results (notably Proposition 2). Some questions also remained regarding the experiments and the fairness of the comparison to existing work.

**Justification For Why Not Higher Score:**

Limited theoretical contribution, open questions regarding the experiments and the fairness of the comparison to existing work.

**Justification For Why Not Lower Score:**

N/A

---

### Decision · Program_Chairs · 2024-01-16

Reject